# Resistance to Targeted Therapy and RASSF1A Loss in Melanoma: What Are We Missing?

**DOI:** 10.3390/ijms22105115

**Published:** 2021-05-12

**Authors:** Stephanie McKenna, Lucía García-Gutiérrez

**Affiliations:** Systems Biology Ireland, School of Medicine, University College Dublin, Belfield, Dublin 4, Ireland; stephanie.mc-kenna@ucdconnect.ie

**Keywords:** melanoma, targeted therapy, tumour suppressor, RASSF1A, resistance, DNA methylation

## Abstract

Melanoma is one of the most aggressive forms of skin cancer and is therapeutically challenging, considering its high mutation rate. Following the development of therapies to target BRAF, the most frequently found mutation in melanoma, promising therapeutic responses were observed. While mono- and combination therapies to target the MAPK cascade did induce a therapeutic response in BRAF-mutated melanomas, the development of resistance to MAPK-targeted therapies remains a challenge for a high proportion of patients. Resistance mechanisms are varied and can be categorised as intrinsic, acquired, and adaptive. RASSF1A is a tumour suppressor that plays an integral role in the maintenance of cellular homeostasis as a central signalling hub. RASSF1A tumour suppressor activity is commonly lost in melanoma, mainly by aberrant promoter hypermethylation. RASSF1A loss could be associated with several mechanisms of resistance to MAPK inhibition considering that most of the signalling pathways that RASSF1A controls are found to be altered targeted therapy resistant melanomas. Herein, we discuss resistance mechanisms in detail and the potential role for RASSF1A reactivation to re-sensitise BRAF mutant melanomas to therapy.

## 1. Introduction

Melanoma is the most common and aggressive form of skin cancer which develops from the uncontrolled growth of melanocytes. The common mole (nevus) is not cancerous but when this cluster of melanocytes within the epidermis changes (dysplastic nevus) or is obviously different in appearance to the common mole, there is a potential for melanoma to develop [1,2]. The development of a melanoma from a benign to a dysplastic nevus, as outlined by the Clarke model, occurs in six stages of gradual expansion into the dermis and lymph nodes, the first of which is the acquisition of a driver mutation [3,4,5]. The melanoma lesion will exhibit any number of the A, B, C, D, E (asymmetry, irregular borders, change in colour, diameter >6 mm, evolution of the tumour over time) characteristics [1].

Melanomas are commonly found on the trunk and face of male patients and the arms and lower legs of female patients but have the potential to develop on all parts of the body, particularly those exposed to UV radiation from the sun [4,6]. Risk factors for the development of melanoma include exposure to UV radiation, phenotypic traits such as fair skin, hair and eye colour, geographical location and a high number of benign nevi [7,8]. In particular, the incidence of melanoma increases when considering the majorly fair skinned populations of Norway, Sweden, the Netherlands, the UK and Ireland who generally have lower levels of melanin to protect melanocytes from mutagenic UV radiation [9]. Familial genetic predisposition is uncommon, though inherited mutations in either *CDKN2A*, a cell cycle inhibitor, or the NER (nucleotide excision repair) pathway, responsible for the DNA damage repair response, have been observed [2,10,11,12]. Families that carry a *CDKN2A* mutation tend to have a higher number of (benign) nevi [12].

Metastatic melanoma (MM) is a highly mutated type of cancer and can be classified into four groups depending on the driver mutation observed—mutation in BRAF (~50% of melanomas), NRAS (~30%), NF1 (10–15%) or triple wildtype [3,13,14]. Other somatic driver mutations can occur in pathways that influence cell growth, metabolism, and cell cycle progression; PTEN, KIT and TP53 [9]. The three most commonly mutated genes converge on hyperactivation of the MAPK pathway. A mutation in BRAF (most commonly BRAF^V600E^) leads to constitutively active RAF, enhancing subsequent MEK and ERK phosphorylation to promote proliferation and prevent cell death [15,16,17]. Less common BRAF mutations also observed include V600K, V600R, V600M [3,15]. NRAS mutations (most commonly in codon 61 but also in codon 12 and 13) result in the constitutive activation of PI3K and MAPK pathways, promoting cell proliferation, differentiation and tumorigenesis [3]. Melanomas with NRAS mutations are often more aggressive and patients have a poorer prognosis [15,18]. Under basal conditions, NF1 inhibits MAPK signalling and so in melanomas with a loss of function mutation in NF1, uncontrolled MAPK pathway activity can lead to tumorigenesis. UV radiation can also stimulate hyperactive MAPK signalling and lead to tumorigenesis [19].

## 2. Treatment of Melanoma

In the early stages (I–III), melanoma can be successfully treated with the surgical removal of the tumour and surrounding tissue [20]. Post-surgery chemotherapy and immunotherapy can be used to ensure the complete eradication of the tumour [20]. In patients who present with tumour ulceration/tumour thickness >0.8 mm, sentinel lymph node biopsy and removal can also be carried out [10,21]. Upon entering the metastatic stages (IV–V), surgery alone is not curative and effective treatment options rapidly deteriorate. The most common sites of metastasis for melanoma are the lung, brain, liver, and intestine [22,23]. Historically, chemotherapy was used as adjunctive therapy for post-surgery patients with MM with only a partial response and modest increase in patient survival [9,21,24]. With the most common mutations in MM converging on the MAPK pathway, targeted therapies have been developed to try to counteract tumorigenic signalling. Therapies which target NRAS as a single agent or in combination with PD-1 inhibitors are being investigated, though to date, mutant NRAS has been a poor therapeutic target [15,25,26].

Considering the high incidence of BRAF-mutated MM, several immunotherapies that target BRAF have been developed, such as vemurafenib, dabrafenib and encorafenib [9,15,21]. While BRAF inhibitors showed promising results with improved patient outcomes [27,28,29,30], a subset of patients did not respond to treatment and the majority of responders rapidly developed resistance and experienced recurrent disease [9,10,21,31,32]. Upon the discovery that BRAF inhibitors alone could induce the paradoxical activation of the MAPK pathway in BRAF wild type cells, MEK inhibitors were developed as a potential combination therapy [32,33]. Clinical trial results indicated that combination therapy did significantly improve patient outcome compared to monotherapy [30,31,32], though the development of resistance persists [34].

Immunotherapies have emerged as a mechanism of activating and engaging the immune system to target tumour cells. Immune checkpoint inhibitors targeting PD1 (nivolumab and pembrolizumab) and CTLA4 (ipilimumab) have been approved for the treatment of melanoma with promising improvements in patient survival [12,35,36,37]. Unfortunately, there are a subset of patients who do not respond to immunotherapy or develop resistance and recurrent disease [10]. The use of radiotherapy in combination with immunotherapy for melanoma treatment has been recently discussed as a potential strategy to enhance immunotherapy response [38]. Moreover, BRAF inhibition seems to enhance radiotherapy sensitivity in melanoma, though several side effects related to skin lesions have been reported when using this combinatory approach [39,40].

## 3. Resistance Mechanisms to Targeted Therapy in Melanoma

The emergence of resistance to BRAF-targeted therapy remains a significant challenge for patients with MM (Table 1). The mechanisms which underly the development of this resistance are varied and can be generally classified into three subtypes: intrinsic, acquired, and adaptive resistance (Figure 1).

### 3.1. Intrinsic Resistance

Considering that patients who have the BRAF^V600E^ mutation maintain disease progression or rapidly develop resistance upon treatment with BRAF inhibitors, at least a proportion of cells within the tumour maintain intrinsic resistance mechanisms [17,34,63]. Intrinsic resistance arises from pre-existing genetic alterations in the tumour or surrounding stromal cells. The molecular mechanisms responsible for the development of intrinsic resistance include increases in proliferative PI3K/AKT and MAPK pathway signalling and disruption to cell cycle regulation [17,34,63,64].

#### 3.1.1. PI3K/AKT Pathway Activation

Loss of phosphatase and tensin homolog (PTEN) tumour suppressor activity leads to the hyperactivation of MAPK and PI3K/AKT signalling. PTEN is a major negative regulator of PI3K signalling which is commonly disrupted by deletion or mutation in melanoma [17,34,44]. In BRAF mutant melanoma, PTEN loss activates MAPK and PI3K signalling through the suppression of BIM-mediated apoptosis, conferring resistance to BRAF inhibitors [34,65,66].

#### 3.1.2. MAPK Pathway Activation

Activating mutations in RAC1 and MEK1 promote the activation of ERK signalling, downstream of BRAF [34]. RAC1 is a Rho GTPase family member which has shown to be mutated (RAC^P29S^) in 4% of melanomas [16,67]. MEK1/2 kinases promote ERK phosphorylation as part of the MAPK signalling pathway and the activating mutation MEK^C121S^ has been shown to increase MEK kinase activity, promoting resistance to both BRAF and MEK inhibitors [16,34,68]. Similarly, the loss of neurofibromin 1 (NF1) observed in BRAF mutant tumour cells leads to intrinsic resistance through the loss of NF1 inhibition of RAS and MAPK signalling [32,37,66].

#### 3.1.3. Disruptions to the Cell Cycle

Cyclin D1 is an important regulator of the cell cycle that promotes cell cycle progression via the inhibition of the RB (retinoblastoma protein) pathway and has been identified as a proto-oncogene [34,69]. *CCND1*, which encodes for the Cyclin D1, protein has been shown to be amplified in BRAF-mutated and wild type melanomas [42]. CDK4 mutations have been observed to increase Cyclin D1 signalling when occurring concurrently with *CCND1* amplification [70]. RB pathway activity has also been diminished in melanoma due to inactivating mutations and the epigenetic silencing of *CDKN2A* [34,70].

### 3.2. Adaptive Resistance

Adaptive resistance mechanisms are acquired by tumour cells to compensate for the loss of BRAF signalling in response to BRAF inhibitors. These mechanisms enhance the survival capacity of the tumour cells to promote proliferation. Adaptive resistance is developed early in response to BRAF inhibition, is reversible [34,71] and generally leads to the re-activation of ERK signalling, upregulation of receptor tyrosine kinases (RTKs), metabolic rewiring, secretion of pro-survival factors by the tumour micro-environment and alteration of gene transcription [34,71]. Considering the speed at which adaptive resistance occurs, effective suppression of this mechanism could contribute to achieving a therapeutic response.

#### 3.2.1. ERK Signalling Reactivation

ERK signal reactivation in response to MAPK pathway inhibitors is achieved through the loss of negative feedback signals. BRAF^V600E^ mutant cells maintain ERK-dependent feedback to supress RTK signalling through the inhibition of negative feedback regulators—SPROUTY (SPRY2,4) and DUSP (DUSP4,6) proteins [34,71,72]. Upon the inhibition of MAPK signalling, ERK suppression of these proteins is reduced, removing the negative feedback regulation of RAS activity, reactivating ERK signalling [34,71,73,74,75].

#### 3.2.2. Alteration of Gene Transcription

The alteration of gene transcription through the up/downregulation of transcription factors is a mechanism of adaptive resistance. MITF (microphthalmia-associated transcription factor) plays an integral role in tumour cell differentiation, invasion and survival and has been characterised as an oncogene in 10–20% of patients with melanoma [34,71,76,77,78,79]. In BRAF inhibitor-treated melanoma cells, MITF was differentially expressed to promote proliferation (high MITF to induce MLANA, PMEL, TYRP genes) or invasion (low MITF to induce high AXL, WNT5A, TEAD, JUN expression) [34,80,81,82].

#### 3.2.3. Alteration of RTK Signalling

The differential expression of RTKs can contribute to the development of adaptive resistance [83]. Following FOXD3 sumoylation of *SOX10*, *ERBB3* (aka *HER3*) expression is upregulated in BRAF inhibitor-treated cells, leading to the activation of the AKT pathway [17,84]. The expression of *SOX10*, which is a negative regulator of EGFR expression, can be decreased in BRAF mutant melanoma cells, leading to enhanced EGFR signalling upon the inhibition of BRAF^V600E^ [17,71]. IGFR and PDGFR receptor induction of MAPK, PI3K and SHH (Sonic Hedgehog) signalling has also been shown to contribute to RTK mediation adaptive resistance [17,85,86].

#### 3.2.4. Metabolic Rewiring

Metabolic rewiring is an established hallmark of cancer, and BRAF mutant melanoma cells exhibit a high level of glycolytic activity (the Warburg effect) [34,71]. Upon treatment with MAPK inhibitors, melanoma cells can rewire and reactivate mitochondrial respiration through the activation of MITF oxidative phosphorylation (oxphos) signalling [34,71,87]. An increase in oxphos has also been demonstrated to be MITF-independent, with the upregulation of JARIDB-mediated oxphos dependent on TFAM (transcription factor A, mitochondrial) [71,88]. The metabolic rewiring of BRAF mutant cells also includes the alteration of fatty acid oxidation. Fatty acid transporter CD36 was upregulated in response to BRAF^V600E^ inhibition, increasing the supply of material for the cell structure during proliferation [32,74].

### 3.3. Acquired Resistance

Long term treatment with MAPK pathway inhibitors can lead to the development of acquired resistance through the development of secondary mutations. Acquired resistance generally results in the reactivation of downstream MAPK signalling.

#### 3.3.1. Secondary Mutations in the MAPK Pathway

Primary mutations in RAS and RAF are generally mutually exclusive. Secondary, activating NRAS mutations have been observed in BRAF^V600E^ mutant cells that are resistant to BRAF inhibitors, leading to the reactivation of MAPK signalling through CRAF [17,34,71,89,90]. Mutant NRAS has also been shown to promote BRAF^V600E^ dimerisation and activity, preventing the activity of inhibitors that only target monomeric BRAF^V600E^ [71,91,92]. Additionally, alterations of BRAF^V600E^ such as amplifications and BRAF^V600E^ splice variants have been identified and proved to confer resistance to BRAF inhibitors [50,90].

Different mutations in MEK 1/2 have been observed in MAPK inhibitor-resistant melanoma. Mutations in exon 3 and 6 have been most associated with the development of acquired resistance [34,93]. MEK1 mutations in E203K, Q56P and K57E have been associated with RAF-independent signalling [34,71]. In cells that are BRAF wild type, BRAF inhibitors have been shown to paradoxically activate downstream MEK/ERK signalling through BRAF/BRAF or BRAF/CRAF dimerisation and activation [71,94].

#### 3.3.2. Activation of Non-MAPK Proliferative Signalling

The suppression of ERK signalling can result in hyperactivation of the PI3K/AKT pathway [32,71,90]. Activating mutations in PI3K or AKT can lead to an enhanced proliferative signal [34,71,89]. YAP/TAZ are transcriptional co-activators that can promote oncogenic signalling and survival upon translocation to the nucleus [95,96,97]. YAP and TAZ expression was demonstrated to be enhanced in BRAF inhibitor-treated melanoma cells [71,98].

## 4. The Tumour Suppressor, RASSF1A

RASSF1A is a tumour suppressor with scaffolding ability that regulates cellular homeostasis by the integration of signals arising from many different pathways [99]. It belongs to the RASSF family of proteins, composed of 10 isoforms named RASSF1-10 [100]. Among the RASSF1 isoforms, RASSF1A is the most extensively studied, followed by RASSF1C, which seems to have opposite effects and has been referred as an oncogene [101]. RASSF1A contains different domains along its protein structure, through which it can bind to and recruit different effectors: C1/DAG (diacylglycerol), ATM (ataxia telangiectasia mutated domain), RA domain (Ras association) and SARAH domain (Salvador-Hippo-RASSF) [99,102]. RASSF1A is found to be associated with microtubules, promoting cytoskeleton stability, and protecting the cell from genomic instability [103,104]. Additionally, RASSF1A interaction with microtubules seems to be essential for its tumour suppressor activity [105]. Moreover, RASSF1A localisation is dynamic, changing throughout the cell cycle phases. It is recruited to the centrosome during G2/M to regulate APC/C-CDC20 activity, leading to proper mitosis timing [106,107,108]. RASSF1A can also regulate G1/S phase transition by modulating Cyclin D1 and A2 levels, CDK4/2 activity and promoting p27 and p21 accumulation and p53 pathway activation [108,109,110,111,112,113,114]. RASSF1A induces apoptosis through a variety of molecular mechanisms. First, ATM phosphorylates RASSF1A upon DNA damage, leading to apoptosis [115]. Second, the activation of the death receptor-induced apoptotic pathway leads to RASSF1A interaction with MOAP-1 and the downstream upregulation of anti-apoptotic BCL2 followed by apoptosis [116,117]. Additionally, RASSF1A promotes apoptosis through the activation of the MST pro-apoptotic signalling pathway. Briefly, RASSF1A stabilizes and activates MST kinases, leading to the downstream activation of LATS1/2 kinases which in turn can elicit apoptosis through the MDM2-p53 pathway or the YAP/p73 pro-apoptotic transcription programme [91,118,119]. Moreover, RASSF1A has been reported to modulate YAP oncogenic signalling in an MST-independent manner. Finally, RASSF1A has been shown to regulate cell migration and invasion through the regulation of RHO GTPases [120,121,122].

*RASSF1A* expression has been reported to be lost in most solid tumours and it has been considered one of the most frequently inactivated tumour suppressors in cancer [123,124]. The hypermethylation of *RASSF1A* promoter and loss of heterozygosity account for the major mechanisms through which RASSF1A expression is reduced [125]. Other mechanisms that result in reduced RASSF1A activity, such as the mi-RNA-mediated downregulation of RASSF1A, point mutations or protein destabilization due to deregulated post-translational modifications have been reported (reviewed in [99]). RASSF1A plays a key role in preventing cancer development and progression through regulating apoptosis, cell cycle progression, genome integrity and microtubule dynamics [126]. It is therefore unsurprising that *RASSF1A* promoter methylation status has emerged as a marker of poor prognosis and aggressiveness in many forms of cancer, including melanoma [127,128,129,130]. Additionally, *RASSF1A* loss correlates with a lack of response to treatment and its restoration contributes to re-sensitization to drug response in a variety of cancers. The ability of RASSF1A to control such broad range of cellular processes relies on the large number of proteins that it scaffolds and therefore, RASSF1A constitutes a pivotal signalling hub on which many different signalling networks converge (recently reviewed in [99,102]).

### RASSF1A and Melanoma

Aberrant epigenetic regulation is a major mechanism involved in the initiation, development, and progression of melanoma. The most intensively studied epigenetic alterations is the altered methylation of promoters and histone modifications. Other, less extensively studied events include chromatin remodelling, positioning of nucleosomes and non-coding RNA-mediated regulation of gene expression. Several tumour suppressor genes have been reported to be silenced in melanoma through CpG-hypermethylation, leading to dysregulation of the cell cycle, signal transduction, DNA repair processes and cell death [131]. *RASSF1A* is among the most frequently reported hypermethylated genes in melanoma, alongside *RAR-β2*, *CDKN2A* and *MGMT* [131]. RASSF1A epigenetic silencing has been reported in more than 50% of melanomas and is considered a predictor of disease progression and patient prognosis [127,128]. *RASSF1A* hypermethylation is found in both melanoma cell lines and tumours, though not in normal skin, benign nevi, or healthy donors, and correlates with a lack of RASSF1A expression or reduced mRNA levels [132,133,134]. Moreover, the re-expression of RASSF1A in BRAF^V600E^-driven melanoma cells has been reported to enhance apoptosis and inhibit tumorigenic potential [135]. Interestingly, BRAF^V600E^ has been associated with CpG-hypermethylation in melanoma and some mechanisms of BRAF^V600E^-mediated promoter hypermethylation have been proposed [136]. These mechanisms include the up-regulation of DMNT1 by BRAF^V600E^, and the phosphorylation and further stabilization of the transcriptional corepressor MAFG, which in turn recruits a complex containing DNMT3 leading to CpG-hypermethylation (reviewed in [99]. Although a direct mechanism involving BRAF^V600E^ in the epigenetic silencing of RASSF1A has never been reported, RASSF1A hypermethylation has been found alongside BRAF (and NRAS) mutations, suggesting a synergistic effect of MAPK pathway mutations and the loss of RASSF1A on melanoma growth [133].

## 5. Potential Impact of RASSF1A Loss on BRAFi Targeted Therapy Resistance

The molecular mechanisms through which RASSF1A exhibits its anti-antitumour activity by regulating different cellular processes have been recently reviewed elsewhere [99,102]. Here, we will describe the molecular pathways deregulated in melanoma, conferring resistance to BRAF inhibition which could also be affected by the loss of RASSF1A. Additionally, we will discuss how the re-expression of RASSF1A could compensate for the lack of other tumour suppressors such as PTEN or CDKN2A and the molecular mechanisms involved. Figure 2 shows a schematic representation of the molecular mechanism described below.

### 5.1. MAPK Pathway

Most of the biochemical mechanisms underlying BRAFi resistance, whether acquired upon prolonged treatment or present at the onset of melanoma development, involve the hyperactivation of the MAPK pathway. These alterations can occur directly, impacting the core components of the MAPK cascade (RAS, RAF, MEK proteins), or indirectly, within upstream regulators of the pathway (i.e., RTKs overexpression, *NF1* loss, PI3K/AKT) [83,137]. Both scenarios lead to alternative mechanisms through which melanoma cells can bypass BRAF inhibition to promote cell survival and melanoma progression. RASSF1A senses oncogenic signalling downstream of RAS, preventing MAPK hyperactivation and pro-survival activity, mainly through the MST pathway. Thus, the loss of RASS1A scaffolding activity could potentially contribute to BRAFi resistance acquisition. Among the mechanisms promoting BRAFi resistance that affect the MAPK signalling pathway, several lead to the dysregulation of RAF1 activity: hyperactivation of RTKs, loss of NF1, mutations in NRAS, increased expression of RAF1, RAF1-BRAF heterodimers, and loss of ERK negative feed-back loops [34,137,138]. Therefore, being able to effectively target RAF1 in cells which do not respond to BRAF inhibition could potentially result in a successful alternative therapeutic strategy. RASSF1A modulates RAF1 activity downstream of RAS [99,102]. Despite its well-known activity as a core kinase within the MAPK pathway, RAF1 plays a key role in preventing apoptosis in a kinase-independent fashion, which cannot be compensated by its homologues BRAF or ARAF [139]. This ability of RAF1 to modulate apoptosis relies on different mechanisms, including the inhibition of MST2 pro-apoptotic signalling, in which the PI3K/AKT pathway and RASSF1A also play key roles, and the inhibition of pro-apoptotic ASK1, an upstream regulator of JNK and p38 MAPK signalling [140].

RAF1 binds MST2 and inhibits its activation by preventing MST2 homodimerization and autophosphorylation [139]. Upon a pro-apoptotic signal, RASSF1A competes with RAF1 for MST2 binding, leading to RAF1-MST2 disruption. RASSF1A induces MST2 autophosphorylation and activation, leading to LATS1 phosphorylation which, in turn, activates the YAP1-p73 pro-apoptotic transcriptional programme [118]. While RAF1-MST2 binding is RAF1 kinase-independent [139], the AKT-mediated phosphorylation of MST2 regulates the RAF1-MST2-RASSF1A axis [141]. Briefly, AKT inhibits MST2 activation by a dual mechanism involving (i) RAF1-MST2 complex formation (preventing RASSF1A-MST2 interaction) and (ii) impairing MST2 autophosphorylation and dimerization (needed for MST2 activation). Therefore, the pro-apoptotic signalling downstream of RAF1 mediated by RASSF1A is tightly controlled by AKT activity. Of the different RAS isoforms, KRAS seems to be the major regulator upstream of RASSF1A that modulates apoptosis through RAF1-MST2 interaction dynamics, while NRAS and HRAS have shown little or no impact [142]. A recent study has shown that K-RAS could be a promising target for melanoma treatment since its inhibition enhances BRAFi-mediated cell death and remains effective upon the development of BRAFi-acquired resistance [143]. In fact, KRAS levels were found to be upregulated upon BRAFi resistance acquisition, leading to enhanced pro-survival AKT and ERK signalling [143]. Considering that RASSF1A can promote apoptosis upon sustained KRAS activation, it would be worth investigating whether melanomas that rely on KRAS to survive could benefit from RASSF1A restoration to promote cell death. However, whether RASSF1A signalling can occur downstream of NRAS or HRAS needs to be further investigated, as RASSF1A activation can be cell-type-specific and context-dependent. Additionally, despite reports that RAF1, but not BRAF, was able to interact with and impair MST2 proapoptotic signalling [139], further studies revealed that BRAF^V600E^ could interact with and inhibit MST1 in thyroid carcinoma [144]. Therefore, investigating whether BRAF^V600E^ competes for MST binding in melanoma and the potential impact of RASSF1A on the restoration of MST apoptotic signalling would be of great interest. On the other hand, RASSF1A expression was found to induce apoptosis in BRAF^V600E^-driven melanoma cells through a mechanism involving ASK1 and p38 MAPK activation [135]. This observation, together with the evidence showing the RAF-mediated inhibition of ASK1 and the interplay between RASSF1A and the MAPK signalling, could suggest another mechanism through which RASSF1A could counteract RAF1 anti-apoptotic activity in melanoma.

### 5.2. PTEN Loss and PI3K/AKT Hyperactivation

The tumour suppressor PTEN, which is an essential upstream regulator of the PI3K/AKT signalling pathway, has been found to be deregulated in many cancer types including melanoma. Loss of PTEN activity is associated with a poor prognosis and increased resistance to treatment. Endogenous PTEN mutations are often found in BRAF driven melanomas and at least 10% of PTEN-null BRAF melanomas show intrinsic resistance to BRAF inhibitors. Additionally, mutations in PI3K and AKT which lead to an hyperactivation of the pathway have been described to contribute to resistance to targeted therapy in BRAF melanomas [34,64,65].

The loss of PTEN promotes the activation of the MAPK and PI3K/AKT pathways, regardless of BRAF inhibition using targeted therapy. RASSF1A integrates the crosstalk of many signalling pathways, including MAKP and PI3K/AKT. In particular, RASSF1A has been described to inhibit PI3K by the direct inhibition of AKT through MST kinases [113,145]. First, RASSF1A interaction with MST1 prevents PP2A-mediated MST1 de-phosphorylation, leading to increased MST1 stability and activation [146,147] which further inhibits AKT activation [148]. Next, RASSF1A-mediated MST1 phosphorylation can result in caspase 3 activation, leading to apoptosis and enhanced MST activity by caspase 3-mediated cleavage of MST1, followed by chromosome condensation and DNA fragmentation [146]. Cleaved MST1 has been shown to further constrain AKT signalling through an independent mechanism than that of the full-length protein, by cleaved MST–AKT direct interaction [148]. In a more recent study, the loss of RASSF1A expression correlated with increased AKT and RAL-GEF signalling in RAS-driven lung cancer [149]. Finally, AKT can bind and inhibit MST2 pro-apoptotic signalling upon PTEN loss. RASSF1A counteracts this effect by competing with AKT for MST2 binding, promoting MST2-mediated apoptosis through c-Jun and p38 kinases [141]. Thus, melanomas showing a loss of PTEN activity which do not respond to BRAFi-based targeted therapy, or which have developed resistance through the hyperactivation of the PI3K signalling pathway could benefit from RASSF1A expression through its direct inhibition of the PI3K/AKT anti-apoptotic signalling.

### 5.3. RAC1 Hyperactivation

The third most common mutation found in cutaneous melanoma occurs within the small GTPase RAC1, in which the P29S mutation induces a constitutively active form of the RAC1 protein [16,67]. The co-occurrence of RAC1P^29S^ and BRAF^V600E^ has been associated with intrinsic resistance to BRAFi and with a mesenchymal-like state of melanocytes [45,59,62]. RAC1^P29S^ activates PAK1, AKT and an SRF/MRTF transcriptional programme involving WAVE-mediated F-actin polymerization promoting a switch from a melanocyte to a mesenchymal phenotype, conferring resistance to apoptosis and survival [62]. There is strong evidence for the inverse correlation between RASSF1A levels and RAC1 activation during cancer progression. The loss of RASSF1A is observed with increased RAC1 activity [126]. Additionally, RASSF1A can impair the activity of oncogenic Rho A GTPase [122] while upregulating Rho-B signalling—the non-oncogenic member of the family [120,121,150]. Although RAC1^P29S^ is no longer regulated by upstream signalling due to its constitutively active RAC1-GTP-like state, a lack of RASSF1A could contribute to the oncogenic potential of hyperactive RAC1 mutant by inducing the loss of the AKT-negative regulation of RAC1 and low Rho-B signalling. As described above, the RASSF1A-mediated activation of MST kinases inhibits AKT [113,145]. Moreover, RASSF1A promotes Rho-B activity by the activation of GEF-H1 through the regulation of NDR kinase and PP2A phosphatase activity [120,121]. The GEF-H1/Rho-B axis regulates epithelial–mesenchymal transition by impairing YAP/SMAD2 nuclear accumulation which is a *bona fide* hallmark of invasive phenotypes. Thus, RASSF1A could counteract RAC1^P29S^-mediated mesenchymal phenotype switch by controlling AKT and YAP oncogenic activities through different mechanisms. Studying the correlation between the loss of the expression of RASSF1A and the mutation state of RAC1 could expose a new avenue for the restoration of sensitivity to targeted therapy in B-RAF-driven melanomas as an alternative to RAC1 inhibitors since they have not been reported to be therapeutically beneficial.

### 5.4. CCND1 Amplification and CDNK2A Loss

Uncontrolled cell proliferation is one of the hallmarks of cancer and can occur via the dysregulation of genes encoding for Cyclin-CDKs or the CDK inhibitors CKIs. *CCND1* amplifications have been found in 17% of BRAF-driven melanomas (11% in melanoma) and have been associated with poor prognosis and intrinsic resistance to BRAFi-based targeted therapy [34]. On the other hand, the loss of expression of the cell cycle inhibitor *CDKN2A* occurs in melanoma through different mechanisms including deletions, inactivating mutations and epigenetic silencing [34]. The combination of *CCND1* amplification and *CDKN2A* loss or CDK4/6 mutations has been associated with resistance mechanisms and shorter progression free survival in patients with BRAF-driven melanoma treated with BRAFi [42]. Cyclin D1 regulation by RASSF1A is of particular interest as a mechanism to counteract *CCND1* amplification. In fact, RASSF1A downregulation correlates with decreased Cyclin D1 levels [151]. It has been shown that RASSF1A promotes cell cycle arrest by impairing cyclin D1 accumulation [112] through a mechanism involving EWS-mediated translational regulation and that seems to be dependent on proper RB-E2F regulation [108]. Additionally, RASSF1A suppresses JNK expression induced by RAS leading to impaired c-Jun phosphorylation, further Cyclin D1 downregulation [114] and p27 accumulation [152], promoting arrested cell proliferation. Since enhanced Cyclin D activity leads to aberrant proliferation, being able to control checkpoints throughout the cell cycle phases could also counteract this effect. In this regard, RASSF1A is a master regulator of cell proliferation, showing differential cell localization during different cell cycle phases [102]. RASSF1A mediates cell cycle arrest at the G1/S phase transition by regulating Cyclin A2 levels through the activation of p120^E4F^ [109,110]. Additionally, RASSF1A plays a key role in the proper regulation of mitotic progression. RASSF1A promotes cell cycle arrest at the G2/M phase by increasing microtubule stability and the inhibition of the APC/C^CDC20^ complex [107,153] through C19ORF5-mediated RASSF1A recruitment to the centrosome in early mitosis [154]. In a later stage, RASSF1A is phosphorylated by Aurora kinases leading to microtubule destabilization and RASSF1A-cdc20 dissociation, promoting RASSF1A- and Cyclin A-mediated degradation and cell cycle progression [106,155,156].

The loss of the *CDKN2A* locus also cooperates with Cyclin D1 in promoting resistance to BRAF inhibitors in melanoma. This locus encodes for two different unrelated tumour suppressor proteins, p16^INK4a^ and p14^ARF^, which regulate cell cycle progression through CDK inhibition or the p53-MDM2 pathway, respectively. Although mutations on p16^INK4a^ are more common in melanoma, genetic alterations affecting p14^ARF^ often co-occur with p16^INK4a^ mutations [157]. p16^INK4a^ major targets are Cyclin D-CDK4/6 complexes; therefore, the mechanisms through which RASSF1A could compensate p16 loss are likely to be similar to those involving Cyclin D activity regulation. On the other hand, the regulation of p53-MDM2 axis mediated by p14^ARF^ could also be, at least in part, compensated by RASSF1A restoration in those melanomas which retain wild type p53. MDM2 is a E3 ubiquitin ligase which, under normal cell conditions, inhibits p53 transcriptional activity by its constitutive mono-ubiquitination followed by proteasome-mediated degradation. p14^ARF^ induces cell cycle arrest at the G2/M phase and apoptosis by antagonizing the inhibitory effect of MDM2 over p53 and thus, it is considered a tumour suppressor. RASSF1A is known to regulate cell cycle progression at the level of G1/S by promoting MDM2 autoubiquitination, preventing p53 degradation [158]. Furthermore, RASSF1A promotes MST2-LATS1 activation downstream of K-RAS in CRC, leading to LATS1 interaction with and sequestration of MDM2, promoting p53 stabilization and apoptosis [119]. Additionally, in CRC, RASSF1A has been found to induce p21 expression through p53 stabilization and MDM2 degradation [111]. LATS2, which can also be activated by RASSF1A, stabilizes p53 levels through the inhibition of MDM2 [159,160]. Finally, ANKRD1, a YAP1 target which is involved in MDM2 destabilization and p53 activation, is upregulated by RASSF1A, antagonizing YAP1 activity [161].

## 6. Restoration of RASSF1A Expression by Demethylating Agents

Genetic alterations, such as deletions, amplifications, point mutations, etc., result in permanent changes within the DNA that are very difficult to repair. In contrast, epigenetic modifications such as methylation or acetylation are not permanent and can be much more easily reversed since they do not affect the DNA sequence directly. Thus, the re-expression of genes which have been silenced by promoter hypermethylation by an epigenetic alteration has been proposed as a promising anticancer strategy. DNA methylation is regulated by DNMT and TET proteins by either transferring or removing methyl groups to CpG islands, respectively [162]. While the hypermethylation of DNA is normally associated with the repression of gene expression, it can also lead to increased promoter activation under specific conditions [163]. Different DNA demethylating agents have been developed throughout the last decade to counteract the cancer-causing effects of hypermethylation [164]. However, the use of DNA demethylating agents can be a double-edged sword considering that they have also been shown to promote the expression of silent oncogenes. Reducing the concentration of demethylating agents or combination with other anticancer therapies are proposed mechanisms for the mitigation of this effect [165]. Of the DNA demethylating agents currently developed, azacytidine and decitabine are approved for the treatment of myelodysplastic syndromes and certain types of leukaemia [166].

RASSF1A promoter hypermethylation has been successfully reverted in different cancer cell models, including melanoma, leading to arrested proliferation, sensitization to different drugs and ultimately, cell death. For instance, RASSF1A expression was restored in a model of lung adenocarcinoma by either treatment with azacytidine or the silencing of DNMT1 expression, leading to reduced proliferation and migration and increased apoptosis [167]. Similar results were observed in mouse models, where the silencing of DNMT1 led to fewer side effects than treatment with azacytidine [167]. Second generation compounds, which are expected to be less aggressive, have been generated and are currently being tested in the clinic. One such compound, SGI-110 (second generation 5-aza-CdR), has been shown to promote RASSF1A expression in ovarian and testicular cancer, promoting apoptosis by sensitizing these cells to cisplatin treatment [168,169]. Similar results have been reported in melanoma. Azacytidine treatment of melanoma cells restored RASSF1A expression [132,133]. A later study also reported that melanomas which had developed resistance to IFN-based treatments, showed the epigenetic silencing of genes by DNA methylation. Moreover, they showed the demethylation of RASSF1A by DNMT1 inhibition using azacytidine-sensitized melanoma cells to IFN treatment [170]. Additionally, a subsequent study in which RASSF1A expression was ectopically restored, revealed that RASSF1A reduced cell viability, cell cycle progression and tumorigenic potential, enhancing apoptosis [135]. Several natural compounds have been shown to exert DNA demethylating activity, promoting RASSF1A re-expression in a variety of cellular models. Although their mechanisms of action are still under investigation, these compounds show promise as potential agents for use in combination with anticancer drugs, as it is thought they would lead to far less toxicity than currently available demethylating agents (reviewed in [99,171]). Altogether, since epigenetic changes can be modulated using external agents, being able to restore the expression of RASSF1A when lost by promoter methylation in melanoma could be considered a promising strategy to overcome resistance to targeted therapy considering the wide range of antitumoral functions that it exhibits.

## 7. Concluding Remarks

Targeted therapy has been a great success for melanoma treatment and has improved patients’ lives in a great extent. However, the main challenge still resides in the resistance mechanism that melanoma cells develop against these treatment strategies. Since tumour suppressors have, by nature, antitumoral activities, being able to restore their functions can be a promising strategy to fight resistance to treatment not only in melanoma, but in every type of cancer.

## Figures and Tables

**Figure 1 ijms-22-05115-f001:**
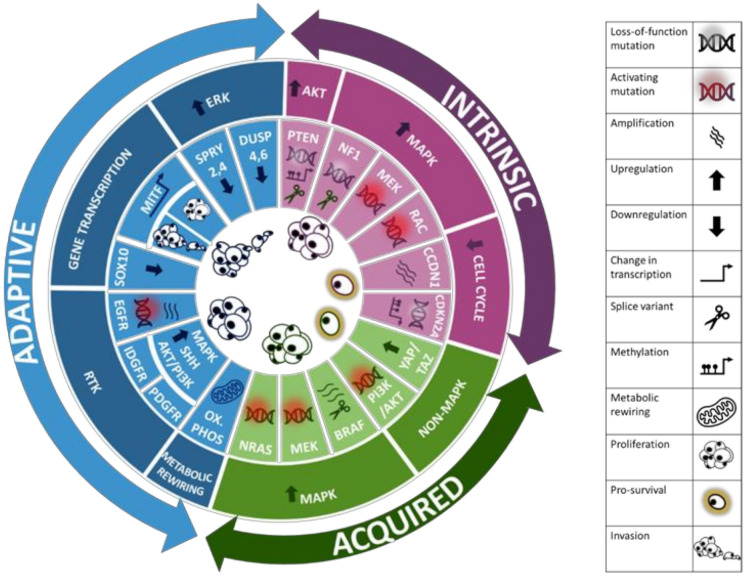
Schematic representation of the three subtypes of resistance mechanism against targeted therapy in melanoma: intrinsic, adaptive, and acquired. The main pathways and mutations accounting for each subtype are displayed.

**Figure 2 ijms-22-05115-f002:**
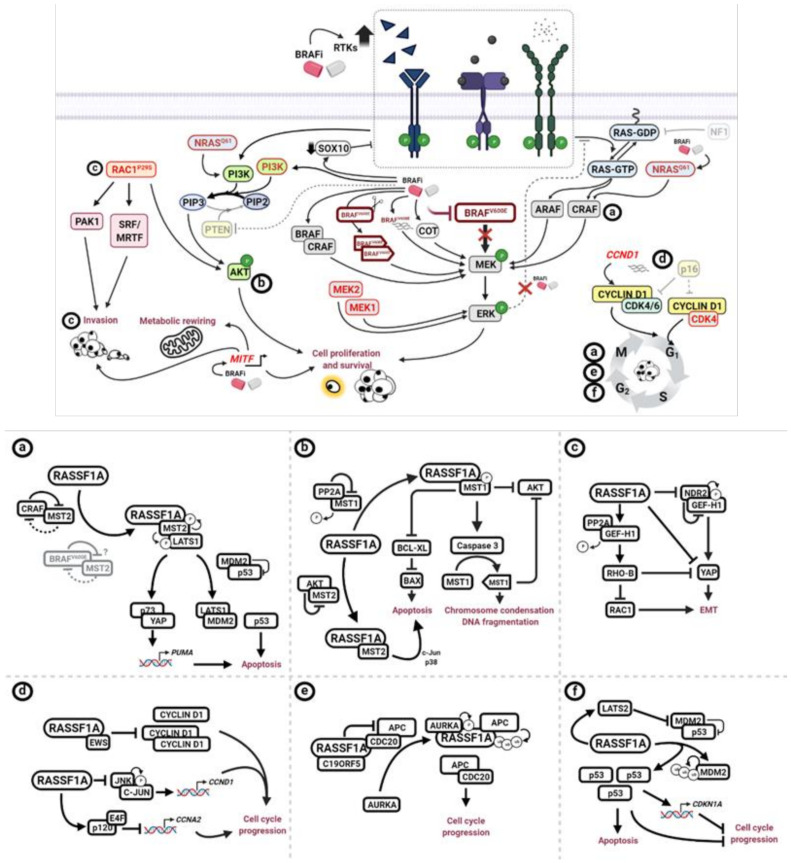
Schematic representation of the molecular mechanisms conferring resistance to BRAFi-targeted therapy (upper panel) and RASSF1A-mediated regulation of associated signalling processes (lower panel). Letters a to f in the upper panel represent the potential impact of RASSF1A on counteracting BRAFi resistance mechanism. Subfigures **a** to **f** from lower panel show the different biochemical mechanism regulated by RASSF1A referred to in the upper panel. (Created with BioRender.com).

**Table 1 ijms-22-05115-t001:** Representative list of mutations contributing to resistance mechanisms to targeted therapy in melanoma. + → resistance; +/− → partial resistance; − → response.

PATHWAY	GENE	ALTERATION	IMPLICATION	Intrinsic	Acquired	Res BRAFi	Res MEKi	Ref
Cell Cycle	*CCND1*	High copy number	Increased expression	+		+		[41]
	*CCND1*	Amplification	Increased expression	+		+		[42]
	CDK4	K22Q, R24C/L	Activating mutation			+	−	[42]
	CDK4	R24C/L	Activating mutation	+/−	+/−	+/−	+/−	[43]
	*CDKN2A*	Low copy number	Low expression	+		+		[41]
	*CDKN2A*	D84N, M53T, N71fs	Inactivating mutation	+	+/−		+	[43]
	*CDKN2A*	deletion, truncation, missense mutation	Loss of function	+/−	+/−	+/−	+/−	[44]
MAPK	NRAS	Missense	Activating mutation		+	+		[45]
	NRAS	Q61	Activating mutation		+	+		[46]
	NRAS	Q61K	Activating mutation		+	+	−	[47]
	CRAF	Increased levels	Increased MAPK signalling		+	+		[48]
	BRAF^V600E^	Amplification	Increased MAPK signalling		+/−	+/−	+/−	[44]
	BRAF^V600E^	High copy number	Increased MAPK signalling		+	+	−	[49]
	BRAF^V600E^	Amplification	Increased MAPK signalling		+	+		[45]
	BRAF^V600E^	Splice variant (p61BRAF^V600E^)	Increased MAPK signalling		+	+		[50]
	MAP2K1	P124SQ/S	Activating mutation	+		+	+	[51]
	MAP2K1	P124S	Activating mutation		+/−	+/−		[44]
	MAP2K1	Q56P	Activating mutation		+/−		+/−	[44]
	MAP2K1	Q56P, E203K	Activating mutation		+	+		[46]
	MAP2K1	P124L	Activating mutation	+			+	[43]
	MAP2K1	V60E, G128V, V154I	Activating mutation	+		+		[45]
	MAP2K1	P124S/L	Activating mutation		+	+		[45]
	MAP2K2	V35M, L46F, C125S N126D	Activating mutation		+	+		[45]
	MAP2K2	W251Ter, A182V		+	+/−	+		[43]
	NF1	X2441_splice	Loss of expression	+		+		[44]
	NF1	Gln282fs, Arg440 *	Loss of function				−	[52]
	NF1	P195S		+		+/−		[43]
	MAP3K8 (COT)	Increased levels	Increased ERK signalling	+	+	+	+	[53]
RTK	EGFR	Amplification, R451C	Increased activity	+	+/−	+/−	+/−	[43]
	EGFR	Demethylation of EGFR regulatory DNA elements	Increased PI3K/AKT signalling		+	+		[54]
	IGF-1R	Increased levels	Increased PI3K/AKT signalling		+	+	−	[55]
	AXL	Increased levels		+	+	+	+	[56]
	KIT	Amplification, G498S	Increased activity	+	+/−	+	+	[43]
	PDGFRβ	Increased levels	Independent MAPK-pro-survival		+	+		[47]
	*SOX10*	Low levels	Increased RTK	−	+	+		[57]
T. microenvironment	HGF	Stromal secretion	Activation of MET	+		+		[58]
PI3K/AKT	PIK3CA	V344G, E545K, H1047R	Activating mutation		+/−	+/−		[44]
	PIK3CA	missense		+	+	+		[45]
	PTEN	mutation, deletion	Loss of function	+/−		+/−		[41]
	PTEN	missense mutation, indel	Loss of function	+		+		[45]
	PTEN	missense mutation, non-sense mutation, deletion, indel	Loss of function		+	+		[45]
	PTEN	Loss/deletion, splice, T27C, P244fs	Loss of function	+	+	+/−	+/−	[43]
	PTEN	deletions, truncation, missense mutation	Loss of function	+	+	+		[44]
Small GTPases (other than RAS)	RAC	P29S	Activating mutation			+	+	[59]
	RAC	P29S	Activating mutation			+	+	[60]
	RAC	P29S/L	Activating mutation	+		+	+	[61]
	RAC	P29S	Activating mutation	+		+	+	[43]
	RAC	P29S	Activating mutation			+		[62]
	RAC	P29S	Activating mutation	+		+		[45]
Metabolic rewiring	*MITF*	Amplification	Increased activity	+	+	+		[45]
	*MITF*	Amplifications, G6R, R316K, S502F	Increased activity	+	+/−	+/−	−	[43]

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
