# Peer review of "Resistance to Targeted Therapy and RASSF1A Loss in Melanoma: What Are We Missing?"

_ijms, 2021, doi:10.3390/ijms22105115_

Round 1

Reviewer 1 Report

The authors have presented a paper about resistance to targeted therapy and rassf1a loss in melanoma.

The topic is of interest for clinicians and researchers involved in the field.

The authors have demostrated a deep knowledge of the subject both in the overall structure of the paper and in the selection of the references.

In my view the paper could be imporved.

In fact there is recent and increaing interest in literature in combination approaches such as Peri-Induction Radiotherapy (PIR) or Post-Escape Radiotherapy (PER) to overcome drug and immuno resistance (see "Immunotherapy and radiotherapy in melanoma: a multidisciplinary comprehensive review, Human Vaccines & Immunotherapeutics, DOI: 10.1080/21645515.2021.1903827").

I would strongly suggest the authors to furhter explore in their manuscript the potential role of radiation therapy in modulating  resistance to targeted therapy.

Reviewer 2 Report

This manuscript by McKenna and García-Gutiérrez the authors provide a systematic review of the molecular mechanisms responsible for drug-resistance in melanoma cells and discuss the roles of tumor suppressor RASSF1A in tumor development and drug response.

Overall this manuscript provides a balanced review of the current understanding of the molecular nature of melanoma tumors. This is an important topic because melanoma is one of the most aggressive forms of cancer that kills millions of people every year, and effective treatment of this disease requires a detailed understanding of its underlying mechanisms. The review by McKenna and García-Gutiérrez is very timely because today the new approaches to melanoma therapy are being actively developed, and one of the most promising strategies at the moment is combining the MAPK-targeting drugs with complementary immunotherapies or additional molecular pathway inhibitors. This paper also highlights the tumor suppressor protein RASSF1A as an important molecular hub involved in cancer proliferation and metastasis regulation whose upregulation can potentially be used as a strategy to mitigate the BRAF therapy resistance.

This manuscript is of a high quality and can be published with a few minor corrections. Those corrections are mostly related to the writing style and do not reduce the value of the paper. Here are my specific minor recommendations:

1)     The authors should provide a more detailed background information about RASSF1A – e.g., explain in the corresponding chapter in more detail what is known about the molecular functions of this protein, its interactors and pathways. This information is summarized in the corresponding figure but this should also be included in the main text to better introduce this protein to the readers who may not be familiar with its molecular activities.

2)     The text needs to be proofread and edited to improve readability and clarity. For example, some sentences are too long and hard to follow, like this one: “Here, we will describe the molecular pathways deregulated in melanoma which confer resistance to BRAF inhibition-based therapy in which the loss of RASSF1A could potentially be directly involved and, the molecular mechanisms in which the re-expression of RASSF1A could compensate for the lack of other tumour suppressor genes such as PTEN or CDKN2A.”. Another example of a sentence that needs editing is this one: “Melanomas are commonly found on the trunk and face of male patients and the upper limbs and lower legs of female patients ”.

3)     The abbreviation MM that authors use throughout the text needs to be explained when it is introduced for the first time.

4)     The authors mention CDKN2A as one of the factors in melanoma. They introduce CDKN2A as a "component of NER (nucleotide excision repair) pathway, responsible for cell cycle repair". First of all, it may be more accurate to say that CDKN2A encodes the cell cycle inhibitors responsible for the suppression of proliferation, rather than for repair per se. And secondly, the authors mention the “cell cycle repair” multiple times throughout the manuscript. It is not clear what they mean by this term. Wouldn’t it be more accurate to say “DNA damage repair” in this case?

Overall, in my opinion, the paper can be published in the journal after these minor points have been addressed.
